Gene expression patterns of novel visual and non-visual opsin families in immature and mature Japanese eel males

Byun Jun-Hwan 1
Hyeon Ji-Yeon 2 3
Kim Eun-Su 2 3
Kim Byeong-Hoon 4
Miyanishi Hiroshi 5
Kagawa Hirohiko 5
Takeuchi Yuki 6
Kim Se-Jae 3
Takemura Akihiro 6
Hur Sung-Pyo hursp@jejunu.ac.kr 2 3
1 Graduate School of Engineering and Science, University of the Ryukyus , Nishihara , Okinawa , Japan
2 Jeju Research Institute, Korea Institute of Ocean Science & Technology , Jeju , Jeju , Republic of Korea
3 Department of Biology, Jeju National University , Jeju , Jeju , Republic of Korea
4 Marine Science Institute, Jeju National University , Jeju , Jeju , Republic of Korea
5 Department of Marine Biology and Environmental Sciences, Faculty of Agriculture, University of Miyazaki , Gakuen–Kibanadai–Nishi , Miyazaki , Japan
6 Department of Chemistry, Biology and Marine Science, Faculty of Science, University of the Ryukyus , Nishihara , Okinawa , Japan
Kim Cheorl-Ho
Electronic publication date: 2020 Feb 27
Publication date: 2020
Volume: 8
Electronic Location ID: e8326
Received 2019 May 19; Accepted 2019 Dec 2
Copyright: ©2020 Byun et al.
Copyright year: 2020
Copyright holder: Byun et al.
License: This is an open access article distributed under the terms of the Creative Commons Attribution License, which permits unrestricted use, distribution, reproduction and adaptation in any medium and for any purpose provided that it is properly attributed. For attribution, the original author(s), title, publication source (PeerJ) and either DOI or URL of the article must be cited.
License URL: https://creativecommons.org/licenses/by/4.0/

Keywords: Photoreceptor, Japanese eel, Anguilla japonica, Opsin, Sex maturation

Funding: National Institute of Fisheries Science R2019003 Ministry of Oceans and Fisheries, Korea National Institute of Fisheries Science R2017038 This work was supported by a grant from the National Institute of Fisheries Science (R2019003) and supported by the project ‘Innovative marine production technology driven by LED–ICT convergence photo-biology’, Ministry of Oceans and Fisheries, Korea, and by a grant from the National Institute of Fisheries Science (R2017038). The funders had no role in study design, data collection and analysis, decision to publish, or preparation of the manuscript.

==============================
This study was carried out to identify and estimate physiological function of a new type of opsin subfamily present in the retina and whole brain tissues of Japanese eel using RNA–Seq transcriptome method. A total of 18 opsin subfamilies were identified through RNA–seq. The visual opsin family included Rh2, SWS2, FWO, DSO, and Exo-Rhod. The non-visual opsin family included four types of melanopsin subfamily (Opn4x1, Opn4x2, Opn4m1, and Opn4m2), peropsin, two types of neuropsin subfamily (Opn5-like, Opn5), Opn3, three types of TMT opsin subfamily (TMT1, 2, 3), VA-opsin, and parapinopsin. In terms of changes in photoreceptor gene expression in the retina of sexually mature and immature male eels, DSO mRNA increased in the maturation group. Analysis of expression of opsin family gene in male eel brain before and after maturation revealed that DSO and SWS2 expression in terms of visual opsin mRNA increased in the sexually mature group. In terms of non-visual opsin mRNA, parapinopsin mRNA increased whereas that of TMT2 decreased in the fore-brain of the sexually mature group. The mRNA for parapinopsin increased in the mid-brain of the sexually mature group, whereas those of TMT1 and TMT3 increased in the hind-brain of the sexually mature group. DSO mRNA also increased in the retina after sexual maturation, and DSO and SWS2 mRNA increased in whole brain part, suggesting that DSO and SWS2 are closely related to sexual maturation.

Introduction

Living organisms recognize various environmental information (light, water temperature, salinity, etc.) according to their specific ecology and have ecologically evolved based on the information. Organisms perceive the external environment through light, transmit the information to the brain, synchronize the biological clock operation to activate the metabolism, and control the physiological and ecological functions by inducing the secretion of endocrine hormones such as melatonin (Benoit, 1978; Falcón, 1999; Falcón et al., 2007; Campbell, Murphy & Suhner, 2001). Mammals have two types of photoreceptor proteins (rhodopsin, cone-opsin) that perceive light in the retina of the eye, and different types of photoreceptors recognize the signals of light (wavelength of light, intensity of light, direction of light, and periodicity) (Hastings & Maywood, 2000; Tada, Altun & Yokoyama, 2009). A photoreceptor is a visual sensory cell capable of recognizing light of a specific wavelength. Photoreceptors also refer to an opsin protein receptor that actually absorbs light and converts it into chemical energy. Vertebrate photoreceptors are regulated by opsin, a superfamily of G-protein-coupled receptor (GPCR), opsin, with an inverse agonist 11–cis retinal chromophore, covalently bound. Indeed, retinal molecules selectively absorb various spectrum of light depending on the formation of binding with the opsin protein (reviewed in Pugh & Lamb, 2000). Absorption of light at a specific wavelength leads to conversion into all-trans form that binds the opsin and transducing proteins, thereby activating a series of visual sensitive-related cellular signal transduction processes (Terakita, 2005). Opsin superfamily is broadly divided into visual and non–visual opsins. There have been extensive studies on vision function of opsin-based photopigment. However, when the opsin was found in tissues such as avian pineal and amphibian skin, opsins were unofficially divided into visual and non–visual groups (Okano, Yoshizawa & Fukada, 1994; Kojima & Fukada, 1999; Van Gelder, 2001). As extra-ocular tissues cannot form images, this classification was suggested. Visual opsin initiates the visual transduction cascade, whereas non-visual opsin is involved in circadian entrainment (Doyle et al., 2008) and retinal metabolism (Bellingham et al., 2003).

The habitats of marine organisms, especially fish, vary in depth and region, ranging from freshwater to brackish areas. The light conditions of these habitats are different in terms of turbidity, color, and brightness (Bowmaker et al., 1994; Bowmaker, 2008). For example, in the case of deep-sea snailfish inhabiting relatively deep-water areas, the spectral sensitivities of the rod and cone photoreceptors react to the blue light (Sakata et al., 2015). In contrast, in the case of black bream, shallow-sea fish, cone photoreceptors (Rh2 or MWS), have maximal light absorbance wavelength (λmax) at 545 to 575 nm, which is the dominant light in their habitat (Shand et al., 2002). Thus, it is presumed that animals have obtained a unique visual system that have made them adapt to the light environment of their habitats in the process of evolution.

The Japanese eel, Anguilla japonica, has been known to have a dynamic life cycle. It migrates to the seawater areas during spawning season and spawns. Sexually immature eels are yellow whereas sexually mature ones are silver. The leptocephalus undergo metamorphosis into glass eels while moving through ocean currents, migrate to freshwater areas where they spend most the time in their life cycle (Tsukamoto, 1992; Tatsukawa, 2003). Thus, the Japanese eel experiences diverse changes in water environment during its life cycle, which differs greatly from other fish and animals.

To date, photoreceptor studies on Anguillid have identified fresh water rhodopsin (FWO) (Zhang et al., 2000), deep-sea rhodopsin (DSO) (Zhang et al., 2000), Rh1d (European eel, A. anguilla and Japanese eel, A. japonica and giant mottled eel, A. marmorata) (Wang et al., 2014), Rh2 (European eel and giant mottled eel) (Cottrill et al., 2009), and SWS2 (European eel and giant mottled eel) (Wang et al., 2014). Molecular biological studies on photo sensitivities of these visual pigments and studies on the expression mechanism of photoreceptors according to ecological stages (glass eel, yellow eel, and silver eel) have been actively conducted. However, the presence or function of a subfamily other than the above–mentioned four types of visual opsin or non-visual opsin subfamily in Anguilla species has not been reported yet. Physiological studies of photoreceptors in vertebrate animals have reported that pinopsin and VA-opsin (Okano, Yoshizawa & Fukada, 1994; Soni & Foster, 1997) in the brain of birds and exo-rhodopsin in the pineal gland of zebrafish directly affect body color change and reproductive physiology (Kojima, Mano & Fukada, 2000; Collin et al., 2009). Thus, it is considered that other types of photoreceptors, except for the previously reported opsins, may play an important reproductive physiological role in Japanese eel but there has been no further investigation into it.

In this study, we investigated the opsin subfamily present in the retina and whole brain tissues of Japanese eel inhabiting Northeast Asia using the RNA-Seq transcriptome. In addition, we examined the opsin subfamily mRNA levels in sexually immature and mature eels using qPCR method. These results identify the physiological role of photoreceptors in the maturation process of Japanese eels and, thus, can be used as a basic material for studies on photoreceptor mechanisms including the effect of environmental factors on maturation and visual adaptation.

Materials & Methods

Experimental fish

We purchased Japanese eels, A. japonica, inhabiting brackish water at Hadori, Gujwaeup, Jejusi, Jeju, South Korea in September at 2016. The wild fish were kept in Lava seawater center in Jeju Techno-park, Jeju, South Korea (33°N, 126°E). The fish were reared for 1 weeks in acryl tank (800 L/capacity) with recirculation system (natural photoperiod = approximately 12L:12D, water temperature 20 ± 1 °C). For the study of the maturation induction of the Japanese eel, males were purchased from an eel aquafarm (Hanwool aquafarm, Gwangju, South Korea). The obtained eels were acclimated in the freshwater round acrylic tank (1 ton/capacity) for at least one week. Light conditions were maintained at 12L:12D using fluorescent bulbs (10W, 600 lx, PPFD = 10.0 µmolm−2s−1, λp = 545 nm) light on at 06:00 and light off at 18:00, and the temperature of the water was maintained at 20  ± 1 °C. All experiments were conducted in compliance with the guidelines of Institutional Animal Care and Experimental Committee of the Jeju National University. The protocol was approved by the Animal care and use committee of the Jeju National University (No. 2016-0039).

For the retina and whole brain RNA-transcriptome analysis (Fig. 1), wild caught Japanese eels (body weight: 233–726 g and body length: 55.3–80.7 cm) were reared for 1 weeks in acryl tank. For the sampling of experimental fish, the retina and the brain were isolated from Japanese eels at 12:00 h and 24:00 h (n = 12, six females and six males) after anesthesia with tricaine methanesulfonate (MS-222, Sigma-Aldrich, ST., USA). The collected tissues were frozen using liquid nitrogen and stored at −80 °C until used for analysis.

Figure 1 Flowchart of the present study.

For maturation artificially induction of Japanese eels, only males (initial body weight: 186.1 ∼227.1 g, n = 6) were selected and reared in the freshwater tank for at least one week. Later, the water was replaced with sea water for one week, and the fish were reared for eight weeks and intraperitoneally injected with human chorionic gonadotropin (n = 6, hCG, 1 IU/g−1) dissolved in saline (150 mM NaCl) at one–week intervals for sexually maturation. During the maturation induction, photoperiod of 12L: 12D (lights on = 07:00, lights off = 19:00) and water temperature of 20 ± 1 °C were maintained, and a complete recirculating aquaculture system (800 L/capacity). Fluorescent bulbs (20W, approximately 600 lx, 10.0 µmolm−2s−1 at 545 nm) were situated above on the tank to provide an illuminance at water surface of 600 lx. After eight weeks of intraperitoneal injection, maturation was determined by the presence or absence of spermiation and histological observation of the testis. For analysis of opsin family genes mRNA level changes in the retina and brain part of Japanese eels before and after maturation, the brain was dissected into the fore–brain, mid-brain, and hind-brain (Fig. 2). The extracted tissues were frozen using liquid nitrogen and stored at −80 °C until used for analysis.

Figure 2 Diagram showing the dorsal view (A) and sagittal plane (B) of the eel brain.

Ob, olfactory bulb; Tel, telencephalon; TeO, optic tectum; Cb, cerebellum; Mo, medulla oblongata; P, pineal gland; SD, saccus dorsalis; PON, preoptic nucleus; SV, saccus vasculosus.

Total RNA isolation and cDNA synthesis

Total RNA was isolated from the retina and three parts of the brain (fore-, mid-, and hind-) using RNA–iso plus (Takara-Bio, Otsu, Japan) according to the manufacturer‘s protocol. After isolated the total RNA, quality, and amount-checked on a 2100 bioanalyzer RNA 6000 NANO chip (Bio-Rad, Hercules, CA, USA) and electrophoresis. cDNA was synthesized using the Transcriptor High Fidelity cDNA Synthesis kit (Roche-diagnostics, Indianapolis, IN, USA) by following the manufacturer‘s protocol.

cDNA library construction and massively parallel sequencing

RNA–Seq paired end libraries were prepared using the Illumina TruSeq RNA Sample Preparation Kit v2 (catalog #RS–122–2001, Illumina, San Diego, CA). Total RNA was isolated from the retina and brain, respectively. After removal of genomic DNA contamination, RNA quality and quantity were assessed by 2100 bioanalyzer RNA 6000 NANO chip (Bio-Rad). High quality total RNA extracted from retina and brain of X individuals were then pooled, respectively. Starting with total RNA, mRNA purified using poly (A) selection was chemically fragmented and converted into single-stranded cDNA using random hexamer priming. Next, the second strand is generated to create double–stranded cDNA. Library construction begins with generation of blunt-end cDNA fragments from ds-cDNA. Then A–base added to the blunt-end in order to make them ready for ligation of sequencing adapters. After the size selection of ligates, the ligated cDNA fragments which contain adapter sequences are enhanced via PCR using adapter specific primers. The library was quantified with KAPA library quantification kit (Kapa biosystems KK4854) following the manufacturer’s instructions. Each library is loaded on Illumina Hiseq2000 platform, and we performed high-throughput sequencing (read length 2 × 100) to ensure that each sample meets the desired average sequencing depth.

Preprocessing and de novo reconstruction of transcriptome

The bases from 5′ end and 3′ end of each read with low quality and adapter sequences were trimmed using Trimmomatic (ver. 0.3.6 Bolger, Lohse & Usadel, 2014), then low averaged quality (Q < 25) were removed by PRINSEQ lite (ver. 0.20.4 Schmieder & Edwards, 2011). Cleaned raw reads from retina and brain RNA were pooled, then mapped to the Anguilla japonica draft genome sequence (Ref) using tophat2 (ver.2.1.0, Kim et al., 2013), then de novo transcriptome reconstruction was performed by a genome-guided Trinity (ver. 2.3.2 Grabherr et al., 2011) with bam mapping result. To remove redundant contigs and create an unigene set, the assembled contigs were clustered and filtered using cd-hit-est with default parameters (CD-HIT package, Li & Godzik, 2006).

Analysis of the opsin DNA sequence

To search opsin family genes from the assembled transcriptome sequences of Japanese eel, the tBlastn program was utilized (E-value < 0.01) on zebrafish opsin protein sequences as queries.

The ORF regions of Japanese eel opsin candidates were found through ORF Finder (http://www.ncbi.nlm.nih.gov/gorf/gorf.html), then presumed protein sequences were aligned with teleost opsin family proteins. A phylogenetic tree was constructed by the maximum-likelihood algorithm using RAxML (Stamatakis, 2014). For quantifying the identified opsin family genes expressions, cleaned reads were mapped on reconstructed contigs by Bowtie2 (Langmead & Salzberg, 2012), then the expression levels were estimated using Tigar2 (Nariai et al., 2014).

Quantitative real-time RT-PCR (qPCR)

Real-time qPCR reactions were performed using the Dice real time thermal cycler (TaKaRa-Bio) and SYBR Premix Ex Taq™ II (TaKaRa-Bio). Gene specific primers used for qPCR were designed using Primer3 plus (Primer Biosoft) and are provided in Table S1. Each PCR reaction mix contained 50% of SYBR Premix, 0.2 µM of each forward and reverse primer, and 2 µl of diluted cDNA template by nuclease-free water. The initial 1 min denaturation was followed by 40 cycles of denaturation for 5 s at 95 °C, annealing and extension for 1 min at 60 °C. To ensure the specificity of the PCR amplicons, the temperature of the sample was gradually raised from 60 to 95 °C as the last step of the PCR reaction and a melting curve analyzed. The primers were successfully tested in the different cDNA samples of the Japanese eel, evaluating that each primer should amplify a single product, reflected as a single peak in the melting curve analysis. The relative mRNA expression levels of target genes were calculated using the ΔΔCt method, and the reference gene was virtually defined as the average of the threshold cycles (Ct) for EF1α.

Histological analysis

The eel testis was fixed in Bouin’s fluid. Fixed testis samples were dehydrated through an ethanol series, embedded in paraffin wax, and sectioned to 7–8 µm thickness. Sectioned tissues were stained with Mayer’s hematoxylin and eosin. State of the sexual maturation was classified into the following 2 stages: immature stage (spermatogonia and spermatocyte; Fig. 3A) and maturation stage (fully spermatozoa; Fig. 3B).

Figure 3 Microphotographs of histological sections of different stages of eel testis and changes of gonadosomatic index (GSI) after hormonally induced sexual maturation.

(A) Immature testis, (B) mature testis, (C) GSI. Scale bar = 200 µm.

Statistics

All statistical analyses were performed using GraphPad Prism 8.0.2 Software. Comparisons of opsin genes expression levels between sexually immature and mature group were performed by the Unparied t test. In the present study, P < 0.05 was accepted as statistically significant.

Results

RNA–Seq transcriptome analysis

Total RNA extracted from the whole brain and retinal tissues of Japanese eels were analyzed using the NGS method. After the adapter trimming and quality filtering, 150,898,925 paired-end reads were survived and used for de novo transcriptome reconstruction. As a result of cd-hit-est clustering, a total of 313,671 contigs (N50 = 965) were obtained. tBlastn and phylogenetic analysis revealed that a total of 18 opsin subfamilies were identified in the retina and the whole brain through RNA–seq (Fig. 4). Among them, the visual opsin families of Japanese eels included rhodopsin2 or middle wave sensitive pigment (Rh2 or MWS), short wavelength-sensitive opsin 2 or blue light sensitive opsin (SWS2), fresh water rhodopsion (FWO), deep–sea water rhodopsin (DSO), and exo-rhodopsion (Exo–Rhod). The non-visual opsin families included four types of melanopsin subfamily (Opn4x1, Opn4x2, Opn4m1, and Opn4m2), peropsin, two types of neuropsin (Opn5-like, Opn5), Opn3 (encephalopsin), three types of teleost multiple tissue opsin (TMT1, TMT2, and TMT3), VA-opsin (vertebrate ancient opsin) and parapinopsin.

Figure 4 Phylogeny of vertebrate visual and non-visual opsins. One thousand bootstrap repetitions were performed and values are shown at the inner nodes.

The zebrafish beta 1 adrenergic receptor was used as an outgroup to root the tree. Analysis was performed with multiple alignments from the amino acid sequence by using ClustalW program. Bold is indicated the visual and non-visual opsin families of Japanese eel, A. japonica.

Changes of GSI

Sexually immature and mature eels were classified based on the histological observation of testis before and after hCG treatment. In the beginning, spermatogonia was mostly observed in the testis of eel males. After eight weeks of hCG injection, spermiation was found in most of male eels, and spermatozoa was mostly observed in lobules. The gonadosomatic index (GSI) was 0.20 ± 0.01 at the beginning and was 25.7 ± 1.4 after maturation, showing a significant difference (P > 0.0001, Fig. 3C).

Changes in opsin family gene expression in the retina between sexually immature and mature eels

Eighteen opsin families identified using the RNA–Seq method were divided into visual opsin (Fig. 5) and non-visual opsin (Fig. 6) families. Then, the mRNA abundance in the retina of sexually immature and mature eels was analyzed using qPCR. In terms of visual opsin expression, the mRNA abundance of DSO increased in the sexually mature group (Fig. 5B), whereas those of FWO and Rh2 were low in the sexually mature group (Figs. 5C and 5D). Non-visual opsin mRNA showed no significant difference between sexually immature and mature groups (Fig. 6).

Figure 5 Visual opsin mRNA level in the retina of sexually immature and mature male Japanese eel.

For the artificially induced sexual maturation, hCG was intraperitoneally injected to the experimental fish group (n = 6) at a concentration of 1 IU/g body weight. Immature fish was sampled before hCG injection (n = 6). Eight weeks after injection, retina was sampled and used for total RNA extraction and cDNA synthesis. The mRNA expression of visual opsin (A–E) in each sample was measured real-time qPCR. Boxplots show min and max values (whiskers), first and third quartiles (box limits), and median (box inner line) of mRNA levels. The asterisk above each bar indicates significant differences according to the unpaired t test (*P < 0.05, **P < 0.01).

Figure 6 Non-visual opsin mRNA level in the retina of sexually immature and mature male Japanese eel.

For the artificially induced sexual maturation, hCG was intraperitoneally injected to the experimental fish group (n = 6) at a concentration of 1 IU/g body weight. Immature fish was sampled before hCG injection (n = 6). Eight weeks after injection, retina was sampled and used for total RNA extraction and cDNA synthesis. The mRNA expression of non-visual opsin (A–M) in each sample was measured real-time qPCR. Boxplots show min and max values (whiskers), first and third quartiles (box limits), and median (box inner line) of mRNA levels. The asterisk above each bar indicates significant differences according to the unpaired t test (*P < 0.05, **P < 0.01).

Changes in opsin family gene expression in brain of sexually immature and mature eels

The brains of the sexually immature and mature eels were dissected into the fore-brain, mid-brain, and hind-brain. Then, the opsin families that showed significant differences were investigated as in the retina (P < 0.05). In terms of visual opsin expression in the brain, DSO and SWS2 mRNA abundance increased in the fore–brain, mid-brain, and hind-brain of the mature group. Other visual opsin mRNAs did not show significant differences in whole brain part (Fig. 7). In terms of non-visual opsin expression in the brain, mRNA abundance of parapinopsin and Opn4m2 increased in the fore brain of the mature group (Fig. 8J), whereas that of TMT2 was low in the mature group (Fig. 9G). Opn4m2 and parapinopsin mRNA abundance increased in the mid-brain (Figs. 8K and 8Q). TMT1 and TMT3 mRNA abundance increased in the hind-brain of the mature group (Figs. 9F and 9L).

Figure 7 Visual opsin mRNA level in the brain of sexually immature and mature male Japanese eel.

For the artificially induced sexual maturation, hCG was intraperitoneally injected to the experimental fish group (n = 6) at a concentration of 1 IU/g body weight. Immature fish was sampled before hCG injection (n = 6). Eight weeks after injection, brain was sampled and used for total RNA extraction and cDNA synthesis. The mRNA expression of visual opsin (A–O) in each sample was measured real-time qPCR. Boxplots show min and max values (whiskers), first and third quartiles (box limits), and median (box inner line) of mRNA levels. The asterisk above each bar indicates significant differences according to the unpaited t test (*P < 0.05, **P < 0.01).

Figure 8 Non-visual opsin mRNA level in the brain of sexually immature and mature male Japanese eel male.

Boxplots show min and max values (whiskers), first and third quartiles (box limits), and median (box inner line) of mRNA levels. For the artificially induced sexual maturation, hCG was intraperitoneally injected to the experimental fish group (n = 6) at a concentration of 1 IU/g body weight. Immature fish was sampled before hCG injection (n = 6). Eight weeks after injection, brain was sampled and used for total RNA extraction and cDNA synthesis. The mRNA expression of non-visual opsin (A–U) in each sample was measured real-time qPCR. Boxplots show min and max values (whiskers), first and third quartiles (box limits), and median (box inner line) of mRNA levels. The asterisk above each bar indicate significant differences according to the unpaired t test (*P < 0.05, **P < 0.01).

Discussion

RNA–Seq transcriptome analysis was performed to examine 18 photoreceptor genes in the retina and whole brain of Japanese eels. As a result, two types of cone opsin (SWS2, Rh2) were identified. However, the presence of a long wavelength–sensitive pigment (LWS) in the long wavelength region was not confirmed in this study. In general, organisms must have at least two cone opsin with different spectra to distinguish colors. Species with one type of cone opsin are considered as “monochromatic vision” or color-blind (Bowmaker et al., 1994). Eels have two or more cone opsin, so they can recognize colors. However, they recognize the wavelength of the narrower region compared with other animals or other fish species. A study suggested that the European eel has two types of cone opsin subfamily, Rh2 (or MWS) and SWS2 cones, so it can distinguish colors (Cottrill et al., 2009). However, the giant mottled eel showed only one type of cone cell that detected a limited range of the optical spectrum (λmax) of 500 nm to 535 nm (Wang et al., 2014). Japanese eels are genetically and ecologically similar European eels; thus, it is presumed that they can recognize colors through two types of cone opsin. In addition, Japanese eels are nocturnal fish and have evolved in a way that they have adapted to dark habitat and/or nocturnal habits. Thus, it is considered that their photoreceptors recognizing the light spectrum of the long wavelength band region may have been functionally atrophied or photoreceptor may have not existed. Similarly, species living in deep-sea or those evolved to adapt to dark environments have been reported to have fewer cone opsins (Mas-Riera, 1991; Pankhurst & Conroy, 1987). SWS1 and LWS gene expression levels were higher in fresh water fish than those in fish inhabiting seawater (Lin et al., 2017). This is because in most freshwater environments, most of the ultraviolet rays penetrating the water surface can be recognized by organisms because of low water depth, while the fish living in deep–sea (50 m or more) tend to lose the LWS gene because the long wavelength (red) is not transmitted to deep-sea regions (Lin et al., 2017).

Figure 9 Non-visual opsin mRNA level in the brain of sexually immature and mature male Japanese eel male.

Boxplots show min and max values (whiskers), first and third quartiles (box limits), and median (box inner line) of mRNA levels. For the artificially induced sexual maturation, hCG was intraperitoneally injected to the experimental fish group (n = 6) at a concentration of 1 IU/g body weight. Immature fish was sampled before hCG injection (n = 6). Eight weeks after injection, brain was sampled and used for total RNA extraction and cDNA synthesis. The mRNA expression of non-visual opsin (A–R) in each sample was measured real-time qPCR. Boxplots show min and max values (whiskers), first and third quartiles (box limits), and median (box inner line) of mRNA levels. The asterisk above each bar indicate significant differences according to the unpaired t test (*P < 0.05, **P < 0.01).

Japanese eel, which was investigated in this study, live in a shallow freshwater region during most of their life span except for spawning. This eel will migrate to deep-sea area of at least 100 m (Cottrill et al., 2009; Tsukamoto, 1992). Regarding this, they may share some genetic characteristics with the deep-sea fish and in this study cone opsin was predicted to be one of the possible genes. The distribution and physiological function of cone opsin appear to be different depending on the level of ecological evolution. In Anguillid sp., only four types opsins (DSO, FWO, Rh2 and SWS2) (Cottrill et al., 2009; Zhang et al., 2000) were studied. Therefore, it is necessary to carry out additional molecular biological and biochemical studies on the range of recognition of color in eels. In this study, opsin families were identified through RNA–seq method, and then highly expressed genes in the retina and brain were analyzed by qPCR. As a result, Parapinopsin mRNA was predominantly expressed in the whole brain, but peropsin and Opn5 were relatively highly expressed in the retina. According to previous studies on opsin expression, the expression of rhodopsin genes in the retina and brain in the ayu (Masuda et al., 2003; Minamoto & Shimizu, 2003), Atlantic salmon (Philp et al., 2000), Japanese eel (Zhang et al., 2000), and percomorph fishes (Cortesi et al., 2015), showed different photoreceptor types and expression sites. However, only the limited physiological function of opsin has been reported.

Non-visual opsin was named in the 1990s, and it has been known to affect circadian rhythms in mammals, reproduction in birds, light avoidance in amphibian larvae, and neural development during egg development in fish (Beaudry et al., 2017). A study on non-visual opsin showed Opn4 expression in the retinal ganglion in mammals, but Opn4 gene was expressed in the retina, brain and skin in non-mammals (Bellingham et al., 2006). In addition, VA-opsin is known to be expressed in the hypothalamus and gonads in birds and fish, and it directly stimulates GnRH in the hypothalamus by recognizing wavelength changes due to photoperiod changes (Davies, Hankins & Foster, 2010; Grone et al., 2007). TMT opsin is expressed in most tissues and embryos in the case of zebrafish. In particular, TMT opsin is expressed in cell lines associated with light entrain able clock (Moutsaki et al., 2003).

In this study, DSO expression increased, and FWO, Opn4m2, VA–opsin, SWS2 and Rh2 expression decreased in the retina during sexual maturation. All of the three brain areas showed the increased DSO and SWS2 expressions. Consistent with the results of this study, a previous study on the photoreceptors of the Japanese eel, reported that DSO expression increased and FWO expression decreased in silver eels (Zhang et al., 2000). In the case of European eel, DSO expression also increased to make eels adapt to the environment before the spawning migration in the early sexual maturation stage. In the late sexual maturation stage, European eels enter deep-sea beyond 100 m depth to spawn, thereby showing a decrease in FWO expression (Zhang et al., 2000). In addition, Japanese conger eel changes its habitat environment from fresh water to open sea while moving from juvenile stage to sexual maturation. To adapt to the changed environment, FWO was mainly expressed in the retina during the juvenile stage, and then DSO expression started to be increased during sexual maturation (Zhang et al., 2002). This may have resulted from Japanese conger eel’s adaptation to the environmental change related to light in the process of migrating to the spawning ground. Analysis of opsin family gene mRNA levels in the Japanese eel brain before and after maturation showed that DSO and SWS2 expressions increased after maturation in all three areas of brain. In addition, DSO expression increased in the retina after maturation, suggesting that DSO is closely related to maturation. However, it is unclear whether the increase in DSO and SWS2 expressions in the brain affects maturation.

In recent years, there have been some studies on the reproductive physiological function of VA-opsin belonging to the non-visual opsin (deep brain photoreceptor) family in the brain. VA opsin was cloned from Atlantic salmon (Soni & Foster, 1997), and VA-long (VAL-opsin) was discovered in zebra fish (Kojima, Mano & Fukada, 2000). Immunohistochemistry studies on Atlantic salmon have reported the existence of opsin-like protein in the hypothalamic nucleus magnocellularis preopticus, suggesting its potential gonadal development control function. There have been few studies on non-visual opsin in the brain, especially its relevance to gonadal development. However, photoperiod action did not influence the gonadal development in ayu without both eyes and pineal gland (Suzuki, 1975), and opsin immune-positive fibers passing through basal hypothalamus were observed in the hypothalamus of Atlantic salmon (Philp et al., 2000). These results suggest that the opsin present in the brain directly affects gonadal development. The level of expression of GnRH located at the top of the BPG axis directly affects reproduction. It is unclear as to whether VA-opsin regulates the expression level of GnRH in the hypothalamus of Japanese quail, but VA-opsin, which affected GnRH expression, was identified in GnRH cells (García-Fernández et al., 2015). Other opsin subfamilies other than VA-opsin may also affect the reproduction system in the hypothalamus, but more research is needed to investigate this hypothesis.

Conclusions

Thousands of opsins have been identified and are divided into eight groups (Terakita, 2005; Yau & Hardie, 2009; Peirson, Halford & Foster, 2009; Terakita, Kawano-Yamashita & Koyanagi, 2012). The current data set shows the diversity of opsins in the animal kingdom because the whole genome sequence is determined in many animals. However, there has been a lack of information on the physiological functions other than the molecular structure or biochemical signals in the retina. In particular, the study of ecologically unique species such as Japanese eels is considered as very important in terms of evolution. In this study, 18 types of opsins were identified in the brain and retina of Japanese eels of which 14 types were new opsin genes. Expression of opsins mRNA in the brain and retina was variable; SWS2 expression was high in all areas of the brain of the sexually mature eels, and TMT3 expression significantly increased in hind-brain. These results suggest that SWS-related shortwave region is directly related to the maturation of Japanese eels. However, follow-up studies are required to demonstrate the relevance. Japanese eels have very unique ecological characteristics, as mentioned above. Unlike other fish species, eco–physiological studies on Japanese eels are necessary to induce artificial maturation through environmental control (light, water temperature etc.), and various studies on the photosensitivity should be continuously carried out.

Supplemental Information

Table S1 Primer sets in this study

Click here for additional data file.

File S1 The result of real-time PCR in the brain

Click here for additional data file.

File S2 The result of real-time PCR in the retina

Click here for additional data file.

File S3 Gonadosomatic index

Click here for additional data file.

The authors are thankful to Chulhong Oh, Moonjeong Lee, Soojin Heo, Dohyung Kang of the Korea Institute of Ocean Science and Technology (KIOST) for their expert assistance and helpful suggestions.

Additional Information and Declarations

Competing Interests

Author Contributions

Animal Ethics

Data Availability

The authors declare there are no competing interests

Jun-Hwan Byun and Ji-Yeon Hyeon conceived and designed the experiments, performed the experiments, analyzed the data, prepared figures and/or tables, authored or reviewed drafts of the paper, and approved the final draft.

Eun-Su Kim conceived and designed the experiments, performed the experiments, prepared figures and/or tables, and approved the final draft.

Byeong-Hoon Kim analyzed the data, prepared figures and/or tables, and approved the final draft.

Hiroshi Miyanishi and Hirohiko Kagawa analyzed the data, authored or reviewed drafts of the paper, and approved the final draft.

Yuki Takeuchi conceived and designed the experiments, performed the experiments, authored or reviewed drafts of the paper, and approved the final draft.

Se-Jae Kim and Akihiro Takemura conceived and designed the experiments, analyzed the data, authored or reviewed drafts of the paper, and approved the final draft.

Sung-Pyo Hur analyzed the data, prepared figures and/or tables, authored or reviewed drafts of the paper, and approved the final draft.

The following information was supplied relating to ethical approvals (i.e., approving body and any reference numbers):

The Animal care and use committee of the Jeju National University approved the study (No. 2016-0039).

The following information was supplied regarding data availability:

The raw data are available in the Supplemental Files.

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
