# Peer review of "Gene expression patterns of novel visual and non-visual opsin families in immature and mature Japanese eel males"

_PeerJ, doi:10.7717/peerj.8326_

## Round 0.1 · original submission · Major Revisions

Your manuscript has some merits and I think that your manuscript can be eventually accepted in our journal. However, I would like to ask for you to carefully revise it. For example, some raw data can be moved to supplementary results.

Reviewer 1 ·

Basic reporting

The authors reported the identification of opsin subfamilies in the retina and the whole brain tissue using RNA-seq. In addition, they examined the expression of these genes in sexually mature and immature male eels by qPCR. This article is very confusing both in how the experiments were conducted and how the data was presented and analyzed. They solely presented all their analyses without any emphasis or deep analysis. Their RNA-seq data did not provide much useful information and is lack of confirmation by qPCR. Therefore, this article is not ready to be published with my concerns listed below.

Experimental design

First, it is vague about how many samples/bio reps were sequenced and how did they pool the samples. The authors mentioned that they had 500m reads from RNA-seq. Was this a combination of samples from both brain and retinal tissues? I also did not find the raw sequencing data for this experiment. Therefore it is hard to evaluate this work statistically.

Second, how they identified and categorized the 18 subfamilies is a mystery. They mentioned that they constructed a phylogenetic tree and performed some kind of blast. However, they did not explain in detail how they recognize and categorize each subfamily.

Third, in the methods, the authors mentioned that they downloaded reference transcripts as the reference for mapping (L178). However, in the results section, they said they de novo assembled contigs. Therefore I was confused about what was used as the reference genome in mapping.

Fourth, for DEG analysis, it is unclear how the statistical test was performed. The authors mentioned using unpaired t-test (L230). However, it is more appropriate to use the adjusted p-value in DESeq.

Fifth, it is odd that the authors used a pie chart to illustrate gene expression levels. It is unclear how the percentage is generated using different bio reps. In addition, they only selected a proportion of genes and they calculated the percentage, it would cause bias in examining gene expression level and makes it hard for comparison. I would recommend the authors to use the normalized counts and generate a heatmap.

Validity of the findings

No comment.

Additional comments

No comment.

Reviewer 2 ·

Basic reporting

This article is clear and unambigous. Only few details need to be improved:
L131: Maybe it is not necessary to repeat
L360: Please, correct immunohistochemical
Paragraph L312: Too much therefore, however, additionaly, ... This paragraph is unclear.

Experimental design

Please add more details in the methods described:
L137: How long did you replace freshwater by see water?
L211: What was the cDNA dilution use?

Validity of the findings

No comment

Additional comments

I consider this paper of high scientific quality. Results could contribute to broaden the knowledge about eel reproductive process and could help in developing strategies to improve their culture. Researchers proposed adequate techniques and research methods to reach their objectives and appropriate means for analyzing the results.

Reviewer 3 ·

Basic reporting

This article is good to meet the topic.

The NGS rawdata and statistic value should be put in the supplements and/or online database before it can be published.

Figure 3. I can’t identify the OTUs and values. It needs to redraw with the large size before it can be published.

Experimental design

It will be better to have a flowchat demonstrated the experimental design and NGS analysis.

Validity of the findings

The NGS rawdata and related statistic value should be put in the supplements and/or online database before it can be published.

---

## Round 0.2 · Minor Revisions

Thank you for the revision. However, there are several points raised. You can easily amend the issues and annotate.

Thanks a lot
Cheorl-Ho Kim

Reviewer 1 ·

Basic reporting

The authors have addressed most of my concerns. However, their Figure 4 is the duplication of Figure 5 which should be corrected. Meanwhile, I would suggest Figure 1,2 and table 1 could be moved to supplemental.

Experimental design

No comment

Validity of the findings

No comment

Reviewer 3 ·

Basic reporting

This manuscript is improved. However, there are still some typos and need to be checked carefully before it can be published.

1. There are typos in literature, for example:
Lin JY, Wang FY, Li WH, Wang TZ >> Lin JJ, Wang FY, Li WH, Wang TY
Check and correct all the literatures again before it is published.

2. Figure 4 is not correct and I cannot see it.
Figure 4 and Figure 5 are the same one?

Experimental design

The experimental design is fine in this version.

Validity of the findings

Figure 4 is not right.

---

## Round 0.3 · accepted · Accept

Dear Dr Hur

Thank you for your revision and rearrangement of the figures, as our collegues pointed out during reviewing.

I am satisfied with the revision and pleased to inform that your submission is now acceptable for publication.

Thank you
Sincerely

Cheorl-Ho Kim
Editor. The Peer J
Prof of Sungkyunkwan Univ (SKKU)